# Evaluation of Climate Change Impacts on the Potential Distribution of Wild Radish in East Asia

**DOI:** 10.3390/plants12183187

**Published:** 2023-09-06

**Authors:** Qingxiang Han, Ye Liu, Hongsheng Jiang, Xietian Chen, Huizhe Feng

**Affiliations:** 1College of Life Sciences, Zaozhuang University, Zaozhuang 277160, China; qingxianghan@163.com; 2Key Laboratory of Aquatic Botany and Watershed Ecology, Wuhan Botanical Garden, Chinese Academy of Sciences, Wuhan 430074, China; jhs@wbgcas.cn; 3School of Environmental Studies, University of Geosciences (Wuhan), Wuhan 430078, China; liuye1119@foxmail.com; 4Wuhan Britain-China School, Wuhan 430030, China; chenxietian2023@163.com

**Keywords:** climate change, coastal plant, MaxEnt modeling, wild radish, potentially suitable habitat

## Abstract

Climate change can exert a considerable influence on the geographic distribution of many taxa, including coastal plants and populations of some plant species closely related to those used as agricultural crops. East Asian wild radish, *Raphanus raphanistrum* subsp. *sativus*, is an annual coastal plant that is a wild relative of the cultivated radish (*R. sativus*). It has served as source of genetic material that has been helpful to develop and improve the quality and yield of radish crops. To assess the impact of climate change on wild radish in East Asia, we analyzed its distribution at different periods using the maximum entropy model (MaxEnt). The results indicated that the precipitation of the driest month (bio14) and precipitation seasonality (bio15) were the two most dominant environmental factors that affected the geographical distribution of wild radish in East Asia. The total potential area suitable for wild radish is 102.5574 × 10^4^ km^2^, mainly located along the seacoasts of southern China, Korea, and the Japanese archipelago. Compared with its current distribution regions, the potentially suitable areas for wild radish in the 2070s will further increase and expand northwards in Japan, especially on the sand beach habitats of Hokkaido. This research reveals the spatiotemporal changes for the coastal plant wild radish under global warming and simultaneously provides a vital scientific basis for effective utilization and germplasm innovation for radish cultivars to achieve sustainable agriculture development.

## 1. Introduction

The effects of climate change are a concern all over the world due to the unprecedented occurrence of extreme weather events, especially with regard to temperature and rainfall [1]. The Sixth Assessment Report (AR6) proclaimed that the global temperature rise could reach 1.5 °C or could be at risk of temporarily breaching it [2]. The AR6 stresses that the increasing severity of extreme climate events has caused the death and disappearance of plants over many local regions [3]. Thus, scores of plant species have been forced to migrate to new areas to exploit more suitable habitats because of climate change. Due to the thermal expansion of waters and the melting of glaciers and ice sheets caused by global warming, the sea level is rising and this has a strong influence on coastal plants. Because of their distinct linear distributions, coastal plants are easily affected by the formation of land bridges during glacial periods. In addition, further climate change is projected to greatly affect crop plants in various aspects, including productivity, growing season, crop variety, and tolerance thresholds to environmental factors [4], and will result in severe impacts on the global crop yield. During domestication, crops go through a genetic bottleneck and end up with much less genetic variation than wild species [5]; thus, they are more likely to be more susceptible to and could be more significantly impacted by climate change. A more variable environment could benefit the wild relatives of crops in the selection of specific desirable traits such as resistance to abiotic stresses and their ability to survive and adapt well [6]. Thus, the wild relatives of plant species used in agriculture continue to serve as a rich source of genetic material, providing a broad pool of potentially valuable genetic resources to improve agricultural production and maintain sustainable agroecosystems [7,8].

East Asian wild radish, *Raphanus raphanistrum* subsp. *sativus* (L.) Domin [9,10], formerly known as *R. sativus* var. *hortensis* f. *raphanistroides* Makino, is a member of the Brassicaceae family and is widely distributed along the coast of East Asia [11]. It is an annual coastal plant and a close relative of the cultivated radish (*R. sativus*), which is an important and popular vegetable crop for its fleshy and edible root, which is consumed worldwide [12,13]. Compared with cultivated radish in East Asia, wild radish presents distinct morphological characteristics, including non-fleshy roots, sparsely to densely hairy stems, pink or purple flowers, cylindric to narrowly lanceolate fruit, and non-shattering mature siliques that mostly contain 1–10 seeds, and robust growing habits, especially in mixed sand and clay soil [14]. It is frequently found as a weed on farmlands, waysides, and seashores in Japan and Korea [12]. It is used as an oil crop to extract cooking oil, a cover crop to fertilize the soil, and also as a promising forage crop in the southwest of China, mainly in the Yunnan and Sichuan Provinces [15]. The extracts of wild radish seeds are rich in sulforaphane, which is regarded as a medicinal resource in foods and has an impact on the development of new functional antimicrobial agents [16]. Various reports have further documented the properties of radish for alleviating constipation [17] and having antimicrobial [18,19], anticancer [20], antioxidant [21,22], and anxiety-reducing activities [23]. Furthermore, the root extract of wild radish is expected to be a potential food product that can deaden neuroinflammation in the brain [24]. As a relative of cultivated radish, wild radish is considered to be a germplasm reservoir of valuable traits, including diverse and effective forms of resistance to insects and diseases and tolerance for drought, salinity, and other abiotic stresses [25]. Thus, it is very important to explore the population dynamics and distribution characteristics of East Asian wild radish from the viewpoint of radish crop evolution and further improvement.

Recently, species distribution models have been broadly adopted to evaluate the impact of climate change on the suitable distribution areas of many species [26]. These models construct statistical connections between the geographical distribution of targeted species and their respective habitats’ environmental variables and then project potentially suitable regions and the future distribution of species under different times frames in the context of global climate change [27]. Most species distribution models call for the collection of geographical distribution using presence and absence data, whereas the maximum entropy model (MaxEnt) only requires the presence data and involves species distribution records and environmental variables [28]. MaxEnt can build distribution maps and variable response curves by testing the retained portion of the training data. Plenty of studies have documented that MaxEnt holds excellent predictive ability in simulation and evaluation; thus, it has been predominantly applied to predict the potential distribution of plant species under current and future scenarios [29,30,31].

Exploring the potentially suitable distribution of wild radish on a large scale is helpful to understand its population dynamics under climate change. Thus, it is worthwhile to utilize ecological modeling to predict the suitable distribution of wild radish and its decisive environmental factors in historic, current, and future periods to reveal the spatiotemporal changes of coastal plant distributions under global warming scenarios and to provide a theoretical basis for cultivating and improving radish crops to accomplish sustainable agriculture development. In this study, the objectives were to (1) identify the key environmental factors that have an impact on the geographical distribution of wild radish in East Asia; (2) predict the areas that are suitable as habitats for wild radish in the Last Glacial Maximum (LGM), current, and future periods; and (3) investigate the climate-change-induced temporal and spatial changes of wild radish’s distribution in East Asia.

## 2. Materials and Methods

### 2.1. Species Occurrence Data

Data points documenting the locations of East Asian wild radish (*R. raphanistrum* subsp. *sativus*) occurrence records were extracted based on the following herbarium databases: the Global Biodiversity Information Facility (GBIF, https://www.gbif.org/, accessed on 15 May 2023), the China Virtual Herbarium (CVH, https://www.cvh.ac.cn/, accessed on 15 May 2023), the National Species Information Infrastructure (NSII, http://www.nsii.org.cn/, accessed on 15 May 2023), and National Museum of Nature and Science in Japan (Snet, http://science-net.kahaku.go.jp, accessed on 15 May 2023). We also extensively reviewed the existing literature and carried out field surveys in East Asia to ensure that we were using the most comprehensive geographic distribution information available. The representative plants were dried and stored as herbariums in our laboratory. We applied Google Earth (http://ditu.google.cn/, accessed on 15 May 2023) to determine and verify the latitude and longitude information of each existence record. After removing all duplicate points, those remaining were subjected to carry out the spatial filtering. Accordingly, a single point was mapped into each grid cell (5 × 5 km). Taken together, 152 unrepeated and high-quality geographically referenced occurrence records for wild radish in East Asia were collected (Figure 1).

### 2.2. Environmental Factors

The 19 bioclimatic variables (Table 1) for the LGM (about 22,000 years ago), current (1970–2000), and future (2070s) scenarios were collected from the Worldclim database (version 2.1, http://www.worldclim.org/, accessed on 10 May 2023) [32] with a resolution of 2.5′. Therein, bioclimatic data for the 2070s represent the average values from 2061 to 2080. Pearson’s correlation coefficient was calculated to avoid potential correlations among climatic variables that may affect the prediction accuracy of models. The general circulation model (GCM) predictions under shared socioeconomic pathway (SSP) scenarios that were produced by the Coupled Model Intercomparison Project Phase 6 (CMIP6) of the Intergovernmental Panel on Climate Change (IPCC) were used to estimate future climate change [33]. Additionally, the GCM was the Beijing Climate Center climate system model (BCC-CSM2-MR) that was developed by the National Climate Center [34]. Two scenarios, i.e., SSP1-2.6 (minimum emission hypothesis) and SSP5-8.5 (maximum emission hypothesis), were chosen for use in this study. In detail, SSP1-2.6 is an updated scenario based on the representative concentration pathway (RCP) 2.6, where a low radiative forcing reaches 2.6 W/m^2^ in 2100, whereas SSP5-8.5 represents an extreme scenario in which no policies are applied regarding greenhouse gases and this causes radiative forcing to be 8.5 W/m^2^ in 2100.

### 2.3. MaxEnt Model Accuracy Verification

MaxEnt 3.4.4 was applied to assess the potentially suitable habitat for East Asian wild radish under the LGM, current, and future climate scenarios [35]. The prediction model was established by randomly selecting 75% of the existing points of wild radish as training data and the remaining 25% as test data. The algorithm was run with 1000 iterations and 10 replicates in each training partition and finally average the results. The dominant environmental factors were mainly chosen by a jackknife test [36]. The default values were retained for other parameters [37].

Threshold-independent receiver-operating characteristic analysis (ROC) was conducted to calibrate the model and validate its robustness. The area under ROC curve (AUC) was tested for additional precision. In general, the AUC values range from 0.5 to 1.0, and the scores could be categorized as 0.5–0.6 fail, 0.6–0.7 poor, 0.7–0.8 fair, 0.8–0.9 good, and 0.9–1.0 excellent [38]. The higher the value of AUC (closer to 1.0), the more accurate the performance of the model [28].

### 2.4. Suitable Habitat Grade Classification

We converted the ASC II format output files of the MaxEnt model into raster files using the ArcGIS format conversion tool to obtain the potential habitat suitability areas for wild radish in East Asia. The potential distribution habitat suitability indexes (0–1.0) were then calculated. Here, 0 is a completely unsuitable habitat and the higher the degree of fitness, the higher is the value. Therefore, the fitness index could be classified into four grades [39]: 0–0.2 unsuitable areas, 0.2–0.4 low suitability areas, 0.4–0.6 medium suitability areas, and 0.6–1.0 high suitability areas.

## 3. Results

### 3.1. Evaluation of MaxEnt Model Prediction Accuracy

The accuracy of MaxEnt software in predicting the potentially suitable distribution of wild radish in East Asia was tested by assessing the AUC value. Under the current period, the mean AUC value of 10 replicated runs was 0.941 (Figure 2), indicating a fantastic level of accuracy. The AUC values of the LGM and the two 2070s scenario simulations were 0.928, 0.947 (2070s, SSP1-2.6), and 0.939 (2070s, SSP5-8.5), respectively, suggesting that the prediction outcomes of the MaxEnt model in our study were accurate and suitable (Appendix A).

### 3.2. Evaluation of the Importance of Climatic Variables

The pairwise correlations between 19 climate variables (Table 1) were detected by Pearson’s correlation coefficients (Figure 3). If the correlation coefficient of any two variables was ≥0.75, only the one with a higher rate of contribution was selected for the later models. For example, the correlation between temperature seasonality (bio04) and temperature annual range (bio07) was found to be 0.94. and bio07 displayed a greater contribution in a tentative jackknife test (Appendix A). Thus, we picked bio07 and simultaneously eliminated bio04 for the subsequent analyses. Finally, a total of 10 variables (bold font in Table 1) were retained for the projection of potentially suitable habitats for wild radish.

In the prediction for the current period, the percent contribution of bio14 was as high as 62.18% and the cumulative contribution rose to 82.21% with the inclusion of bio15 and bio02, indicating that these variables best explained the data. The two variables with the highest permutation of importance were those of the minimum temperature of the coldest month (bio06) and the annual range of temperature (bio07), producing a cumulative value of 66.44% (Appendix A). The contribution rate of each environment varies greatly in different periods. However, one thing that they have in common is that bio14 always shows the largest contribution rate (Appendix A), highlighting its unassailable dominance.

The MaxEnt model’s internal jackknife test of environmental variable importance showed that precipitation of the driest month (bio14) and precipitation seasonality (bio15) contributed the most to the model, followed by the mean diurnal range (bio02), mean temperature of the driest quarter (bio09), temperature annual range (bio07), and minimum temperature of the coldest month (bio06) (Figure 4). The cumulative contribution of the above-described six factors is 93.86% (Appendix A). These results collectively showed that precipitation and temperature are the primary environmental factors affecting the geographical distribution of wild radish in East Asia.

Depicting the relationship between environmental variables and species occurrence probability, species response curves exhibit the biological tolerances for target species and their habitat preferences. Based on the species response curves that were constructed, wild radish prefers the precipitation of the driest month (bio14) to range from 40.6464 to 112.8912 mm, precipitation seasonality (bio15) to be <52.1592, mean diurnal range (bio02) to be >3.66 °C, temperature of driest quarter (bio09) to be <11.7251 °C, temperature annual range (bio07) to be <30.4412 °C, minimum temperature of coldest month to be (bio06) >−5.72 °C; this is when the probability of the presence of wild radish was greater than or equal to 0.5 (Figure 5).

### 3.3. Distribution and Change in Potentially Suitable Habitat under Different Climate Conditions

The total suitable area of the current distribution of wild radish was about 102.5574 × 10^4^ km^2^, with the low-, medium- and high-suitability areas comprising about 73.7610 × 10^4^ km^2^, 15.7760 × 10^4^ km^2^, and 13.0204 × 10^4^ km^2^, respectively (Table 2). Guangxi, Guizhou, Hunan, Jiangsu, Zhejiang, Taiwan, the southern region of Korea, and almost all of Japan were determined to be suitable areas for wild radish. The coastal areas of Japan, Korean Jeju Island, and China’s Jiangsu, Zhejiang, and north Taiwan were the highly suitable areas. Guangxi, Guizhou, and Hunan had a few fragmented suitable areas (Figure 6a). From the LGM to the current period, the distribution pattern of suitable habitats of East Asian wild radish showed clear and evident differences that were mainly due to the decline and even disappearance of highly suitable habitats (Figure 6b). In summary, the percentage of high-, medium- and low-suitability areas shrank by approximately 29, 20, and 10 × 10^4^ km^2^, respectively. Southeastern China and the seacoasts of Korea and Japan lost a large quantity of area suitable for sustaining the wild radish population in the wild.

Under the SSP1-2.6 scenario framework in the 2070s, the highly suitable habitat (13.0591 × 10^4^ km^2^) remained almost the same as the current levels; however, the area of low-suitability habitat and the total suitable area in 2070 declined by 5.8334 and 2.7599 × 10^4^ km^2^, respectively (Table 2), indicating a downward trend in preferred habitat availability (Figure 6c). By contrast, in scenario SSP5-8.5, the poorly and moderately suitable habitats locally expanded in central and southern China and their corresponding areas increased by 10.4871 and 13.0479 × 10^4^ km^2^, respectively (Table 2). As a result, the total suitable area increased significantly, reaching approximately 124.96 × 10^4^ km^2^, 21.85% more than the current distribution area. In Hunan, Guizhou, and Guangxi Provinces a large area will change from low to moderate suitability (Figure 6d). Interestingly, the northernmost boundary of the distribution was predicted to move to the north of Hokkaido, Japan’s northernmost island, under each future climate scenario.

## 4. Discussion

Potentially suitable areas for wild radish were predicted based on 152 global occurrence records and 10 environmental variables using the MaxEnt model. The AUC value offers a threshold-independent measure of the overall accuracy of the model and is an important model quality indicator for evaluating the accuracy of the applied model [40,41]. The AUC value of this study under each scenario framework, i.e., LGM, current, and future, is much greater than 0.9, suggesting that the prediction model exhibits an excellent fitting ability. Thus, the results of MaxEnt model prediction can objectively reflect the distribution of wild radish in East Asia.

Through the division of the fitness grade, the distribution maps of the potential suitable areas of wild radish in East Asia under different climatic conditions were obtained. The results showed that the potential distribution areas of wild radish are mainly concentrated in the Taiwan, Zhejiang, Sichuan, Yunnan, Guangxi, and Fujian Provinces of China and the southern border of Korea, as well as the coastal areas of Ryukyu and mainland of Japan. This is essentially in line with its current actual distribution in East Asia [42]. In addition, wild radish, as a newly recorded species, has been discovered in coastal areas in Fujian [43], Shanghai [44], and Wenzhou [45], which is consistent with the potential distribution obtained in this study, indicating the robustness and reliability of the model.

Among the 10 environmental factors used in the model, the total contribution rate of precipitation-related factors was 72.43% and that of temperature-related factors was 27.57%, indicating that precipitation is the primary environmental factor affecting the distribution of wild radish in East Asia. The precipitation of the driest month (bio14) is between 40.6464 to 112.8912 mm, and it was found to be the most predominant variable with a contribution rate of 62.18% indicated by the jackknife test. Precipitation may play a major role in shaping the ecological adaptation of wild radish and has a great impact on its geographical distribution. Previous studies implemented on radish have shown that storage roots, plant height, leaf area, photosynthesis, and seed development were significantly affected when there was a decrease in water availability [46,47]. The precipitation of the driest month was consistently reported to be one of the most significant bioclimatic variables in the control of the distribution of Zingiber species in China [48]. Therefore, the availability of water should be the most critical factor in crop plant growth and dominantly affects its distribution and survival.

Temperature has been reported to affect the vegetative growth [25], flowering time [49], seed dormancy, and germination [50] of wild radish. Among the studied temperature factors, the minimum temperature of the coldest month (bio06) had the largest contribution (11.45%), with the threshold being >−5.72 °C, implying a possible wild radish preference for a cold climate. This is in agreement with previous studies that reported radish is a seed vernalization type of plant and needs a period of cold treatment (about 4 °C) for vernalization to initiate flowering availability [51,52]. In addition, except for the minimum temperature of the coldest month (bio06), other temperature-related factors including mean diurnal range (bio02), isothermality (bio03), maximum temperature of the warmest month (bio05), the annual temperature annual range (bio07), mean temperature of the wettest quarter (bio08), and the mean temperature of the driest quarter (bio09) collectively contributed 16.13% to the distribution model for wild radish. This suggests that the survival and growth of this species was less sensitive to the temperature of the habitat than rainfall. This relatively wide temperature adaptation range for the ecological niche or strong adaptability to habitat is also reflected in the report in [53], in which vernalization is not absolutely necessary for wild radish populations in southern Japan, where the wild radish exhibits a facultative vernalization requirement.

Wild radish has a wide distribution encompassing southwest China, South Korea, and Japan. Our model indicated that the total suitable habitat area encompassed ca. 102.55 × 10^4^ km^2^ for the species under current climate conditions. During the LGM period, the sea level in East Asia was decreased by about 120–150 m compared with the current sea level, thus Japan was connected to the East Asian continent and formed the East China Sea Shelf [54]. The land bridges had a strong influence on the geographic distribution of plants [55]. In addition to predicting the distribution range of wild radish in East Asia during the LGM, we speculate that it was likely the “corridor” that allowed wild radish to migrate eastward to escape the unsustainable cold and dry climate. Wild radish significantly shifted eastward to the coastal areas in central Japan and the Ryukyu Islands in the south. In addition, in our field survey the natural population of wild radish in Fujian, Guizhou, and Guangxi was found to be sparse and discontinuous. The possible reason for this might be the strong impact of human activities, especially the destruction of coastal habitats.

For year 2070, the suitable range of wild radish was slightly reduced under the RCP 2.6 but greatly increased under the RCP 8.5. This finding is consistent with that of previous studies [29,56], which indicated that temperature had a favorable effect on plants by accelerating phenological processes and extending the growing season. However, other studies have reported that temperature might have a negative effect on crop plants, e.g., potato [57], Zingiber [48], and maize [58]. The impacts that climate change produced on crops may vary among species and this might contribute to their different survival temperature thresholds. Our study found that the presence of wild radish was less sensitive to temperature, implying a strong ecological adaptability. In future climate conditions where microclimatical temperature continues to rise, the predicted temperature along the northern border of Japan would reach the low temperature threshold for wild radish growth, which might give rise to the northward shift of the suitable regional boundary for wild radish to survive. These results are consistent with speculation that, with global warming, plants will expand towards higher latitudes [59].

Apart from increasing temperature, the rising atmospheric CO_2_ concentration will be a fiercely changing environmental factor in the future [60]. From the perspective of CO_2_ concentration, the total suitable distribution area of wild radish in a high CO_2_ concentration (124.9686 × 10^4^) environment will be much larger than that in a low CO_2_ concentration (99.7975 × 10^4^) environment. This finding indicates that an elevated atmospheric CO_2_ concentration will have a positive effect on wild radish growth, which is in agreement with Bhargava and Mitra’s reports on crop plants [61]. Increases in atmospheric CO_2_ concentration are expected to boost photosynthesis and the accumulation of carbohydrates to fertilize plant growth [62]. CO_2_ enrichment in the atmosphere is expected to drive a crop yield increase over time by promoting stomatal closure and saves water [63]. However, studies of maize crops have described that the low positive effect of CO_2_ and adaptation are insufficient to offset the negative effects of increasing temperature and will eventually lead to yield losses [64]. Due to the interactions between increasing CO_2_ concentration and temperature on plant growth being complicated [65], the influences of these two variables on crop yields or coastal plants should not be overlooked or unraveled individually when studying climate change impacts.

## 5. Conclusions

Global climate changes exert a great impact on the distributions of coastal plants and the wild relatives of crops, particularly in East Asian flora. In this study, the MaxEnt model was used to predict the potentially suitable distribution areas of a typical coastal plant, *R. raphanistrum* subsp. *sativus*, generally named wild radish in East Asia, under the LGM, current, and future climatic conditions. Our results showed that the precipitation of the driest month (bio14) and precipitation seasonality (bio15) were the two most dominant environmental factors that affected the geographical distribution of wild radish in East Asia. The predicted distribution area was found to be largest during the LGM period as a result of the decreased sea level. Compared with the current distribution, the suitable range for wild radish will be slightly reduced under the RCP 2.6 but will greatly increase under the RCP 8.5 in the 2070s. In any of the future scenarios, the ecological niche of wild radish is predicted to undergo a small expansion into high latitude areas. This research reveals the spatiotemporal changes of coastal plants under global warming and simultaneously provides a vital scientific basis for the effective utilization and germplasm introduction for radish cultivars to achieve sustainable agriculture development.

## Figures and Tables

**Figure 1 plants-12-03187-f001:**
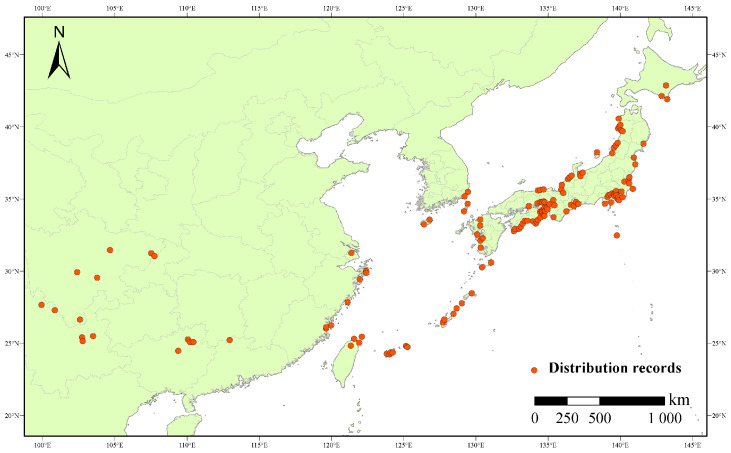
Distribution records for wild radish in East Asia.

**Figure 2 plants-12-03187-f002:**
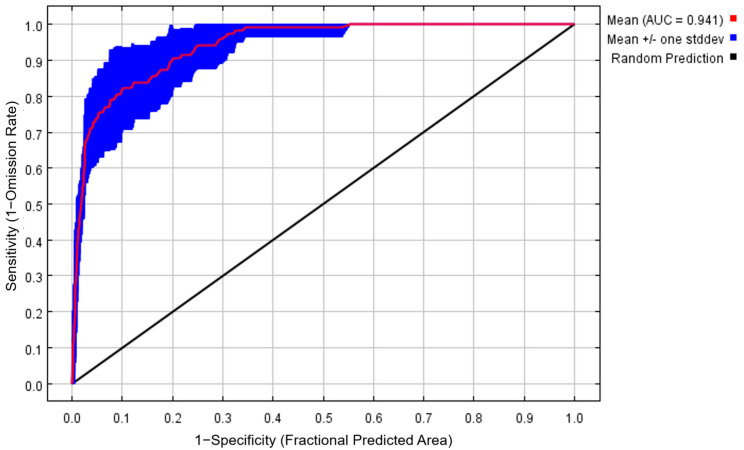
The receiver-operating characteristic (ROC) curve and area under ROC curve (AUC) value under the current period (mean for 10 replicated runs).

**Figure 3 plants-12-03187-f003:**
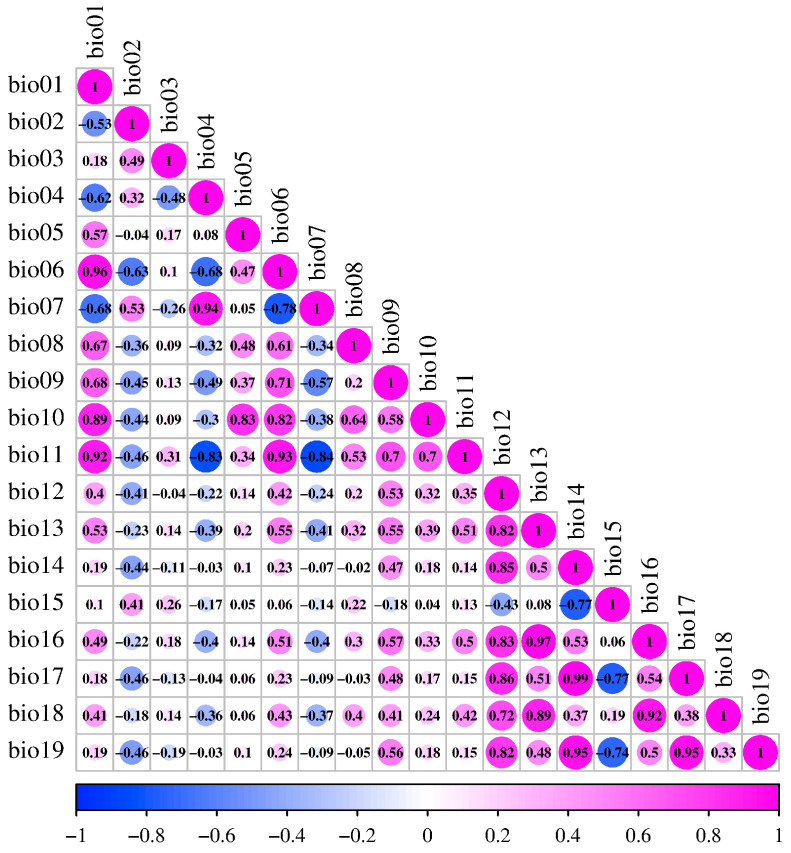
Pearson’s correlation coefficient between climate variables. Interpretations of the abbreviations of climatic variables are listed in Table 1. The values in the circles represent the correlation coefficient |r| and the positive and negative values represent positive and negative correlation, respectively.

**Figure 4 plants-12-03187-f004:**
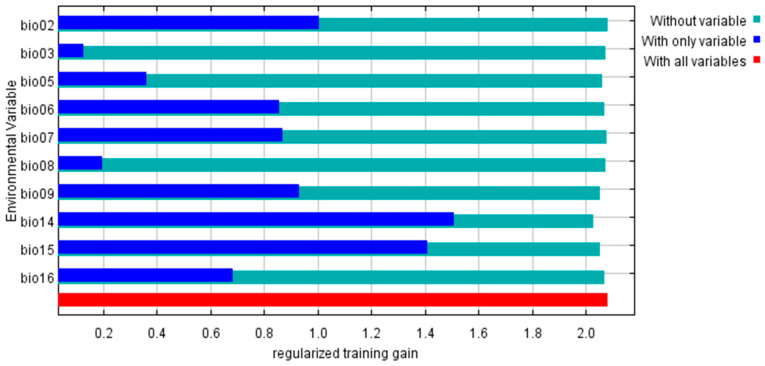
The relative predictive power of each environmental variable based on a jackknife test.

**Figure 5 plants-12-03187-f005:**
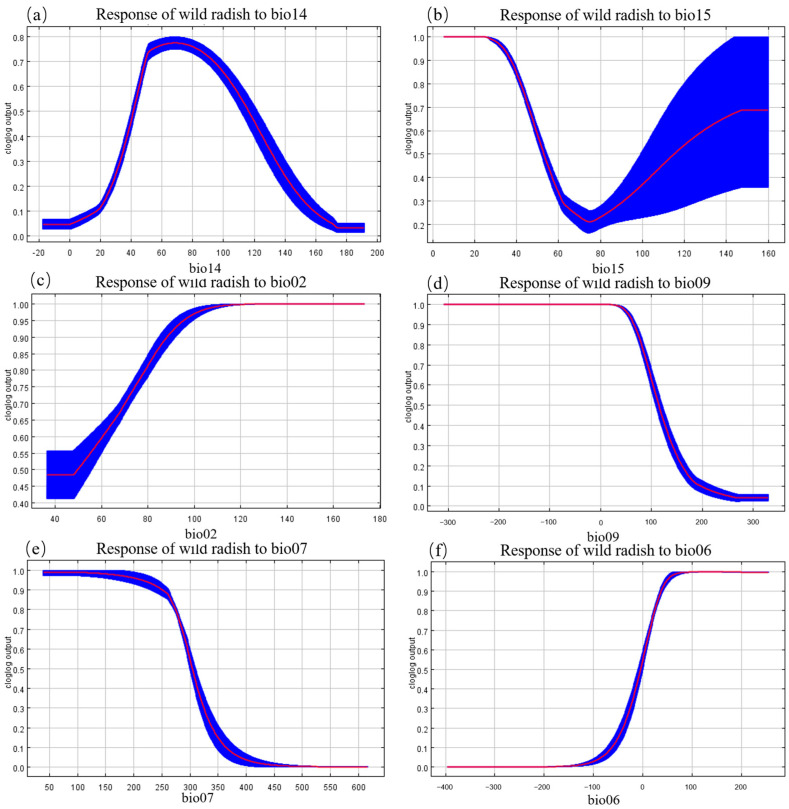
Wild radish response curves in relation to (**a**) precipitation of the driest month (bio14), (**b**) precipitation seasonality (bio15), (**c**) mean diurnal range (bio02), (**d**) temperature of driest quarter (bio09), (**e**) temperature annual range (bio07), and (**f**) min temperature of coldest month (bio06).

**Figure 6 plants-12-03187-f006:**
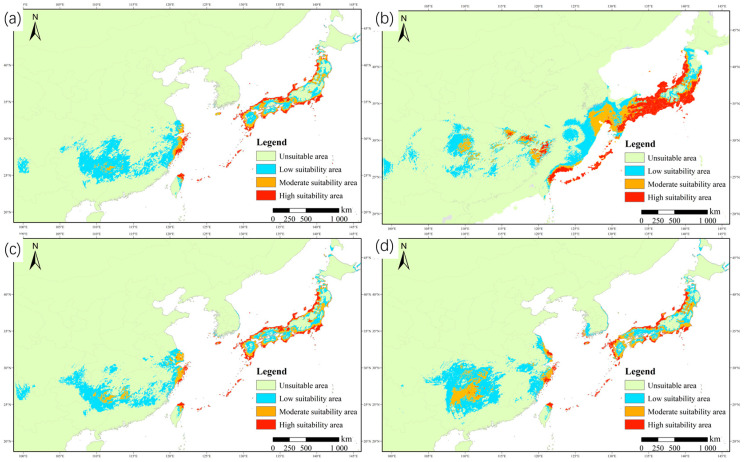
Potential distribution of wild radish in East Asia: (**a**) Current; (**b**) LGM; (**c**) 2070 for shared socio-economic pathway (SSP)1-2.6; (**d**) 2070 for SSP5-8.5.

**Table 1 plants-12-03187-t001:** Description of bioclimatic factors used for preliminary screening and final construction of the maximum entropy model (MaxEnt) model.

Variables	Description	Units
bio01	Annual Mean Temperature	°C
**bio02**	Mean Diurnal Range (Mean of the monthly (maximum–minimum temperatures))	°C
**bio03**	Isothermality (bio02/bio07) (×100)	-
bio04	Temperature Seasonality (Standard Deviation ×100)	-
**bio05**	Maximum Temperature of the Warmest Month	°C
**bio06**	Minimum Temperature of the Coldest Month	°C
**bio07**	Temperature Annual Range (bio05-bio06)	°C
**bio08**	Mean Temperature of the Wettest Quarter	°C
**bio09**	Mean Temperature of the Driest Quarter	°C
bio10	Mean Temperature of the Warmest Quarter	°C
bio11	Mean Temperature of the Coldest Quarter	°C
bio12	Annual Precipitation	mm
bio13	Precipitation of the Wettest Month	mm
**bio14**	Precipitation of the Driest Month	mm
**bio15**	Precipitation Seasonality (Coefficient of Variation)	-
**bio16**	Precipitation of the Wettest Quarter	mm
bio17	Precipitation of the Driest Quarter	mm
bio18	Precipitation of the Warmest Quarter	mm
bio19	Precipitation of the Coldest Quarter	mm

Note: The climatic factors in bold were ultimately used to build the model.

**Table 2 plants-12-03187-t002:** Area of each suitable habitat under the Last Glacial Maximum (LGM), current, and future periods.

Time Period	Area of Each Suitable Habitat (The Change in Area Compared with the Current Period) 10^4^ km^2^			
	Lowly Suitable Habitat	Moderately Suitable Habitat	Highly Suitable Habitat	Total Suitable Habitat
LGM	83.0800 (−9.3200)	35.4804 (−19.7044)	42.2017 (−29.1812)	160.763 (−58.2055)
Current	73.7610 (0.0000)	15.7760 (0.0000)	13.0204 (0.0000)	102.5574 (0.0000)
2070 (SSP1-2.6)	67.9276 (−5.8334)	18.8107 (3.0348)	13.0591 (0.0387)	99.7975 (−2.7599)
2070 (SSP5-8.5)	84.2481 (10.4871)	28.8238 (13.0479)	11.8967 (−1.1238)	124.9686 (22.4112)

## Data Availability

Not applicable.

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
