# Peer review of "Evaluation of Climate Change Impacts on the Potential Distribution of Wild Radish in East Asia"

_plants, 2023, doi:10.3390/plants12183187_

Round 1

Reviewer 1 Report

The article presents the results of research on the impact of climate change on the distribution of wild radish - plants of great importance in East Asia, due to the possibility of extracting cooking oil, medicinal properties, the possibility of using as a cover crop to fertilize the soil, as well as a valuable forage crop.

I find the paper interesting, although it concerns the distribution of only one species, and in addition one that is not a species of outstanding economic or other importance. The basis of the research was the analyse of data on the distribution of wild radish and meteorological data from publicly available databases. Field studies were also carried out. In total, 152 unique geographical records of wild radish in East Asia were collected. On this basis, a model used maximum entropy algorithm was built. Its accuracy was also verified. The data were well analysed, the conclusions result from the conducted research. The manuscript generally was good structured and easy to read. My only suggestion is to modify the chapter order. In my opinion, the methodology should come before the results section.

Reviewer 2 Report

The authors took a widely used MaxEnt model to check the potential changes in distribution of wild radish, based on the available presence data and 2 climate scenarios. A short explanation of the tool and how results are shown is provided, along with the results and used variables description. The discussion briefly explain main outcomes of the analysis with some relevant references. I found the language clear, presentation engaging (especially maps) and overall it is a solid piece of scientific writing. 

However, I am amazed by the order in the article - because a lot of abbreviations are used, materials and methods with all the explanations of the terms, must be presented before the results etc. In another words, before evaluating the model accuracy, one must know how to read the graphs made in the software and what are the coefficients used. Please correct the order and make sure sufficient explanations are provided also for the readers that are not familiar with the tool you used. Examples to be fixed: I am not familiar with the AUC and do not know how the curve was produced in the first place. Other instance is the Pearson's correlation coefficient for the climatic variables which is nowhere to be found i.e. supportive figure is just more content of figure 1.

Also, I would change the second citation and better refer to the newest global IPCC report, to avoid misleading in the sentence in lines 33-34.

Reviewer 3 Report

 Dear Authors:

This is a strong paper and I recommend that it be accepted for publication pending additional editorial and literature consideration.  I have a made some suggested wording changes for consideration in the attached file in which potential edits are indicated in red. This paper contributes excellent original information and predictions regarding the historic and future distribution of wild radish. 

Are there other studies on other species with paired wild and cultivated taxa for which this kind of predictive work has been conducted?  It might be useful to cite some similar efforts in other taxa.

Were plants from your study and study sites vouchered and placed in Herbaria, and if so, in which Herbaria are they preserved and accessible to researchers?

The name Raphanus sativus var. hortensis f. raphanistroides Makino may no longer be valid nomenclature, and it is likely a synonym – or is no longer in use (particularly f. raphanistroides).  If so, this should be mentioned, although as historic nomenclature it has always been associated with the wild radish element of the broader species.  At least one reference referred to var. hortensis as a cultivated/cultivar only taxon.

The concluding sentence in the Abstract suggests that “This research revealed the spatiotemporal changes of coastal plants under global warming, and simultaneously provided vital scientific basis for effective utilization and germplasm innovation for radish cultivars to achieve sustainable agriculture development.”  Is the germplasm of wild populations in the new and projected distribution likely to have changed within the 2070 timeframe?  I’m sure it will over time, but this is a relatively short interval.

It would be useful to cite at least one source providing a detailed botanical description of the plant species – I’ve attached a link for the Jepson Manual’s description for Raphanus sativus, but one that details wild radish/ R. sativus var. hortensis would be good to cite.  If there isn’t an available description of var. hortensis, a description of sativus in the wild radish distribution would be useful.

The following references should be considered for inclusion:

Swaamy, K.R.M. 2023.  Origin, distribution, genetic diversity and breeding of radish (Raphanus sativus L.).  International Journal of Development Research 13 (02):  61657-61673.

CABI  https://books.google.com/books?id=aDw5Tk6ar9sC&pg=PA141&lpg=PA141&dq=Raphanus+sativus+var.+hortensis+f.+raphanistroides+Makino+original+description&source=bl&ots=5z0dIRPNsD&sig=ACfU3U25q3sXN1rL6sdd492oLuwtmCRlFQ&hl=en&sa=X&ved=2ahUKEwjIhLmlktGAAxXTAjQIHX7eC1Y4FBDoAXoECAIQAw#v=onepage&q=Raphanus%20sativus%20var.%20hortensis%20f.%20raphanistroides%20Makino%20original%20description&f=false

Kaneko, Y., C.Kimizuka-Takagi, S.W. Bang and Y. Matsuzawa. Chapter 3. Radish. In Gene Mapping in Molecular Breeding and Molecular Breeding in Plants.  Vegetables.  C. Kole (ed).  2007.  Springer-Verlag 2007.  See page 141 and the two papers cited: “…The wild radish, the so-called Hama-daikon, R. sativus var. hortensis f. raphanistroides Makino (Kitamura 1958), or R. raphanistrum spp. maritimus (Hinata 1955), was naturally grown on the sea coast in East Asia…”

https://powo.science.kew.org/taxon/urn:lsid:ipni.org:names:2876199-4

Prosea https://uses.plantnet-project.org/en/Raphanus_sativus_(PROSEA)

Cultivar group names are proposed here.

·       Cv. group Chinese Radish. Synonyms: Raphanus sativus L. var. niger (Miller) Persoon (1807), var. hortensis Backer (1907), var. longipinnatus Bailey (1923).

The Jepson Herbarium eFlora https://ucjeps.berkeley.edu/interchange/

https://ucjeps.berkeley.edu/eflora/eflora_display.php?tid=40992

https://www.researchgate.net/publication/280870388_Distinct_Phylogeographic_Structures_of_Wild_Radish_Raphanus_sativus_L_var_raphanistroides_Makino_in_Japan

Distinct Phylogeographic Structures of Wild Radish (Raphanus sativus L. var. raphanistroides Makino) in Japan.  PLOS ONE DOI: 10.1371/journal.pone.0135132

15 pages

Abstract and figures

Coastal plants with simple linear distribution ranges along coastlines provide a suitable system for improving our understanding of patterns of intra-specific distributional history and genetic variation. Due to the combination of high seed longevity and high dispersibility of seeds via seawater, we hypothesized that wild radish would poorly represent phylogeographic structure at the local scale. On the other hand, we also hypothesized that wild radish populations might be geographically differentiated, as has been exhibited by their considerable phenotypic variations along the islands of Japan. We conducted nuclear DNA microsatellite loci and chloroplast DNA haplotype analyses for 486 samples and 144 samples, respectively, from 18 populations to investigate the phylogeographic structure of wild radish in Japan. Cluster analysis supported the existence of differential genetic structures between the Ryukyu Islands and mainland Japan populations. A significant strong pattern of isolation by distance and significant evidence of a recent bottleneck were detected. The chloroplast marker analysis resulted in the generation of eight haplotypes, of which two haplotypes (A and B) were broadly distributed in most wild radish populations. High levels of variation in microsatellite loci were identified, whereas cpDNA displayed low levels of genetic diversity within populations. Our results indicate that the Kuroshio Current would have contributed to the sculpting of the phylogeographic structure by shaping genetic gaps between isolated populations. In addition, the Tokara Strait would have created a geographic barrier between the Ryukyu Islands and mainland Japan. Finally, extant habitat disturbances (coastal erosion), migration patterns (linear expansion), and geographic characteristics (small islands and sea currents) have influenced the expansion and historical population dynamics of wild radish. Our study is the first to record the robust phylogeographic structure in wild radish between the Ryukyu Islands and mainland Japan, and might provide new insight into the genetic differentiation of coastal plants across islands.

Kalloo, G. and B.O. Bergh (eds.). 1993.  Genetic Improvement of Vegetable Crops. Pergammon Press, Ltd. https://doi.org/10.1016/B978-0-08-040826-2.50039-4   Publisher Summary

Radish, Raphanus sativus L., is an annual vegetable belonging to the family Cruciferae and is a traditionally important vegetable in many countries. The enlarged root and hypocotyl of radish are consumed mainly as a salted vegetable and are also eaten fresh as grated radish, garnish, and salad. Radish has been cultivated and consumed as a part of the eastern Asian diet. Recent advances in cultivation technique require more promising varieties of radish that can resist the problems of disease and insects and can meet a more varied range of dietary demands. This chapter presents an overview of the germplasm resources and cytogenetics of radish. It discusses the present state of radish breeding, mainly in Japan. For radish, breeding work has been carried out on ecological traits, resistance to diseases, and adaptability for different kinds of consumption. The ecological traits are productive and qualitative characteristics such as high yielding ability, early maturity, late bolting, edible quality (pungency), late pore formation, cold-hardiness, drought resistanceheat tolerance, wet tolerance, soil adaptability, and so on. Virus disease, yellows, soft rot, downy mildew, grey leaf spot, and other diseases are prevalent in Japan. Radish is mainly consumed fresh, boiled, or salted or as dried strips or seedlings (kaiware). Radish is an allogamous plant exhibiting a high level of self-incompatibility and shows inbreeding depression when self-propagation by bud pollination is repeated. It is difficult to obtain a large amount of seed mainly because of the limited seed numbers produced per pod. In this case, F1 hybridization combined with self-incompatibility and heterosis is a helpful breeding method. There are three methods of seed production in F1 hybridization: (1) single crossing, (2) three-way crossing, and (3) double crossing. The chapter describes the double crossing method, which is the most effective means of seed production in radish.

Do, M.H., Kim, M., Choi, SY. et al. 2021. Wild radish (Raphanus sativus var. hortensis f. raphanistroides) root extract protects neuronal cells by inhibiting microglial activation. Appl Biol Chem 64, 31. https://doi.org/10.1186/s13765-021-00604-7

Abstract

External stimulus-induced activation of microglia plays an important role in the protection of neurons in the central nervous system; however, over-activation of microglia could cause neuronal damage, and it is implicated in the pathogenesis of neurodegenerative diseases. The aim of the present study was to investigate the effects of wild radish (Raphanus sativus var. hortensis f. raphanistroides) root extract (WRE) on microglial over-activation. Mouse microglia BV-2 cells and rat primary microglia were stimulated with lipopolysaccharide (LPS), treated with WRE, and analyzed for nitric oxide (NO) production, pro-inflammatory cytokine secretion, inducible NO synthase (iNOS) expression, and p38 kinase phosphorylation. Human neuroblastoma SH-SY5Y cells were treated with microglia-conditioned medium and analyzed for cell viability. Stimulation with LPS increased NO production and iNOS expression in BV-2 cells and primary microglia, but the treatment with WRE decreased both. Furthermore, WRE downregulated the mRNA expression and secretion of inflammatory cytokines interleukin-1 beta (IL-1β) and tumor necrosis factor alpha (TNF-α), and inhibited the phosphorylation of p38 in LPS-activated microglia. Treatment with the conditioned medium of LPS-induced BV-2 cells decreased the viability of SH-SY5Y cells, but the damaging effect was significantly alleviated in cells treated with the conditioned medium of LPS plus WRE-cultured microglia. This indicated that the WRE treatment of microglia could protect neuronal cells from microglial activation-induced neurotoxicity. WRE may be a potential food product to attenuate neuroinflammation via the inhibition of microglial over-activation, which can slow down the neurodegenerative processes in the brain.

Hur, J., Choi, S. Y., & Yeom, M. (2021). Neuroprotective Effect of Wild Radish Extract on Scopolamine Induced Memory Impairment. Journal of the Korean Society of Food Culture, 36(6), 633–639. https://doi.org/10.7318/KJFC/2021.36.6.633

Abstract

Raphanus sativus var. hortensis f. raphanistroides Makino (Korean wild radish [WR]) are root vegetables belonging to the Brassicaceae family. These radish species mostly grow in sea areas in Asia, where they have been traditionally used as a medicinal food to treat various diseases. To investigate the effect of WR on neuronal cell death in SH-SY5Y cells, beta-amyloid was used to develop the cell death model. WR attenuated neuronal cell death in SH-SY5Y and regulated the mitogen-activated protein kinase (MAPK) signaling. WR extract also inhibited acetylcholinesterase inhibitor activity. Additionally, the WR treatment group ameliorated the behavior of the memory-impaired mice in a scopolamine-induced mouse model. In the behavior test, WR treated mice showed shorter escape latency and swimming distance and improved the platform-crossing number and the swimming time within the target quadrant. Furthermore, WR prevented histological loss of neurons in hippocampal CA1 regions induced by scopolamine. This study shows that WR can prevent memory impairment which may be a crucial way for the prevention and treatment of memory dysfunction and neuronal cell death.

iNaturalist – this doesn’t need to cited, but does provide information about the distribution based upon GIS based photographs

https://www.gbif.org/occurrence/search?dataset_key=50c9509d-22c7-4a22-a47d-8c48425ef4a7&taxon_key=6306597

I have introduced a number of suggested wording changes into the narrative your manuscript for your consideration.  I hope that they are helpful and my intent was not to alter the meaning of any of your material.
